# The rhizodynamics robot: Automated imaging system for studying long-term dynamic root growth

Aradhya Rajanala[1]*, Isaiah W. Taylor[2], Erin McCaskey[1], Christopher Pierce[1], Jason Ligon[3], Enes Aydin[1], Carrie Hunner[3], Amanda Carmichael[3], Lauren Eserman[3], Emily E. D. Coffee[3], Anupam Mijar[2], Milan Shah[2], Philip N. Benfey[2], Daniel I. Goldman[1]

1 Department of Physics, Georgia Institute of Technology, Atlanta, GA, United States of America,
2 Department of Biology, Duke University, Durham, NC, United States of America, 3 Atlanta Botanical Garden, Atlanta, GA, United States of America

☺ These authors contributed equally to this work.
* arajanala3@gatech.edu

**Data Availability Statement:** All code files are available at https://github.com/the-rhizodynamics-robot.

## Abstract

The study of plant root growth in real time has been difficult to achieve in an automated, high-throughput, and systematic fashion. Dynamic imaging of plant roots is important in order to discover novel root growth behaviors and to deepen our understanding of how roots interact with their environments. We designed and implemented the Generating Rhizodynamic Observations Over Time (GROOT) robot, an automated, high-throughput imaging system that enables time-lapse imaging of 90 containers of plants and their roots growing in a clear gel medium over the duration of weeks to months. The system uses low-cost, widely available materials. As a proof of concept, we employed GROOT to collect images of root growth of *Oryza sativa*, *Hudsonia montana*, and multiple species of orchids including *Platanthera integrilabia* over six months. Beyond imaging plant roots, our system is highly customizable and can be used to collect time- lapse image data of different container sizes and configurations regardless of what is being imaged, making it applicable to many fields that require longitudinal time-lapse recording.

## Introduction

All organisms must perform effective environmental exploration for survival. Plants are sessile and interact with the location where they grow; therefore, they can only navigate and explore their (typically) below-ground environment through dynamic elongation of their root systems. In contrast, animals move themselves to different locations via a wide variety of locomotory strategies and behaviors. The movement of animals and the strategies used to navigate complex environments have been well documented [1–5]. While recent advances have begun to allow the monitoring of root growth, many challenges remain in studying typical plant systems, which normally grow in opaque environments [6–9]. Insight into growth and movement control principles of both plants and animals in complex environments can have impacts on a wide variety of applications including robotics, healthcare, and conservation efforts [10–12].

**Funding:** This work was funded the National Science Foundation (nsf.gov) grant NSF PHY-1915445 awarded to D.I.G and P.N.B, the Dunn Family Professorship awarded to D.I.G, and the Gordon and Betty Moore Foundation (moore.org) grant GBMF3405 awarded to P.N.B. The funders had no role in study design, data collection and analysis, decision to publish, or preparation of the manuscript.

**Competing interests:** The authors have declared that no competing interests exist.

Each organism presents unique technical problems in the design of an experimental system for studying long time behavior. However, there are some common features of apparatus design that are applicable to the study of both animal and plant behavioral dynamics. For instance, time-lapse imaging has been essential to study complex systems in plant and animal contexts [4, 6, 9]. Beyond time-lapse imaging, understanding organismal behaviors and environmental interactions often requires *longitudinal* recordings of dynamics (over weeks or even months), which presents an additional set of apparatus design criteria. High-throughput time lapse imaging produces data sets that are comparable across organismal types, allowing cross-system comparisons and the development of generic behavioral metrics. A wide range of technology platforms have been designed for the study of animal movement; however, there exist only a few high throughput imaging devices for plants [13–17]. Some time-lapse studies of plants have been conducted at the field scale covering acres of growing crops as well as in controlled experimental laboratory settings. Examples of systems deployed in the field include the RootTracker [18], which uses capacitive changes in below-ground electrodes to determine real-time root architecture for a growing plant. Additionally, the Vinobot and Phenobot [19] can perform high throughput field above ground phenotyping and collect data for 3D image processing. While the RootTracker provides insight into root growth below ground, it cannot provide the detailed root structure data that is achievable through time-lapse imaging.

In contrast to studies of non-fossorial animal movement and the aboveground portions of plants, a major obstacle to studying root growth dynamics is the opacity of soil. While x-rays can help visualize root structures within soil, it is difficult to achieve a high throughput scale through this method. In addition, the use of x-rays is resource intensive [20]. Utilizing transparent gel substrates to enable optical access has helped address this challenge; however, the long time scale of root growth has typically restricted throughput. A few high throughput time-lapse imaging systems have been developed to image roots, including the RhizoTube, which allows simultaneous imaging of shoot and root growth [16], the RootArray, which allows imaging of 16 roots with confocal microscopy [21], and robotic gantry systems developed to image multiple petri dishes of growing Arabidopsis seedlings [15]. These imaging systems have enabled greater understanding of plant root growth but are, in general, system specific, and are limited in their generalizability to multiple plant types and experimental conditions.

To address this problem, we designed a low-cost automatic robotic imaging system, the Generating Rhizodynamic Observations Over Time (GROOT) robot. Our system can be constructed by researchers with limited prior knowledge of robotics and is customizable to the needs of a variety of different applications. Our system is one of few designed to image plant roots in a high-throughput manner and leverages modern technological advances in imaging and automation technology allow greater flexibility and customization than prior systems. In this paper we demonstrate its capabilities through acquisition of a six-month time lapse study of orchid root growth, recording multiple orchids in each position as they fully grow in the span of a month.

## Materials and methods

GROOT consists of two cameras mounted to a multi-tier integrated two-axis translational gantry system (Fig 1G). The cameras are translated and sequentially stopped in front of each growing root system and collect simultaneous images from different views. For experiments requiring only a single side view, one camera is sufficient. A light-proof curtain encloses the entire setup and LED lights are used to control light cycles during plant growth and illuminate plants for imaging. The system is controlled using an Arduino micro-controller and images

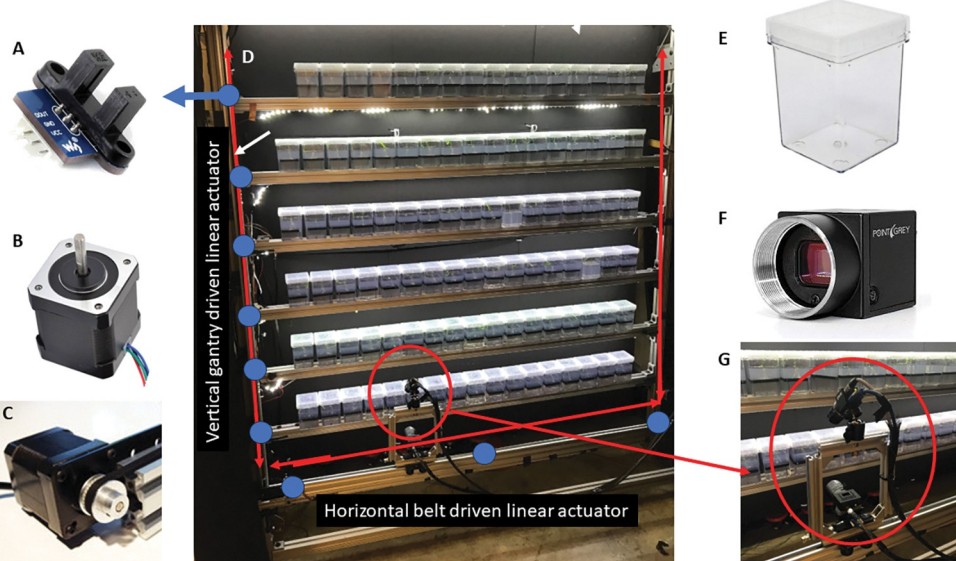

**Fig 1. GROOT design and components.** Generating Rhizodynamic Observations Over Time (GROOT) robot design and components. A) Photointerrupter sensors and their locations (marked with blue dots) on the GROOT system. B) NEMA 17 stepper motor. C) Openbuilds belt driven linear actuator showing NEMA 17 motor attachment with belt over the motor shaft. D) GROOT system built at Georgia Tech showing the vertical and horizontal belt driven linear actuator locations. E) Magenta 7 polycarbonate containers used for imaging. F) FLIR Flea3 Camera with GPIO cord attached to use for hardware trigger functionality. G) Dual camera setup used for GROOT with angled camera positioned above forward-facing camera.

are saved using the software FlyCapture which is compatible with FLIR USB cameras. We describe construction of a simple imaging robot in more detail in S1 File: Robot assembly. The code to operate this robot is described in S2 File: Robot Control (code available at https://github.com/the-rhizodynamics-robot).

In the orchid root observations described below, each growing root system was placed into a container with dimensions 70*70*100 mm. It is possible to quickly remove and replace containers from the gantry setup. Plants were grown in Yoshida's nutrient solution solidified with gellan gum [22].

The frame for the imaging system was built with belt driven linear actuator kits from Openbuilds, which uses Nema 17 stepper motors (Fig 1B) to control movement of the belts (Fig 1C). One horizontal and two vertical kits are set to allow the camera to move along a two-axis frame (front frame). This frame was fastened parallel to a second frame (back frame) made of the same material that has rows with adjustable heights to allow for containers of different sizes. The cameras were mounted to the front frame on the horizontal linear actuator.

LED light strips were attached on the underside of the frame for each row of containers. They were connected to a relay, allowing each row to have its light cycle programmed independently so individual rows can be illuminated during the dark growth cycle for image capture.

While stepper motors are designed to count steps and thus move predetermined distances, we found that over time small motor slips or miscounts caused slight deviations in imaging, necessitating photointerrupter sensors to be placed on the camera. To ensure the camera stops in the same place each time a picture is taken (e.g. in front of a container), photointerrupter sensors are attached at different locations on the front frame (Fig 1A). Each corner of the horizontal frame has a sensor to communicate the start and end points for horizontal movement. There is also a sensor placed in the middle of the horizontal frame as a fiducial mark. When

the camera approaches a sensor, a small trigger piece sends a signal from the sensor to the Arduino microcontroller, which executes the specific code programmed to occur when that signal is received. When the end of a row is imaged, the camera moves to a set position where the photointerrupter sensor is located before moving up or down to the next row. In this way, the camera always starts and ends precisely in the same location as it executes an imaging sequence. Each row also has a sensor placed on the vertical frame so that the camera will stop at the same row height during each cycle, reducing image drift over the large timescales employed here. After several tests in which we determined that limit switches and hall effect sensors would be insufficient for our purposes, photointerrupters were determined to work the most effectively for long term plant imaging application.

The length of an imaging cycle can be set in the code so that when the camera reaches its starting position (bottom left corner of the array) at the end of a cycle, it pauses until the counter for the time period is finished, allowing time-lapse imaging at any interval greater than the time required to traverse the array (~14 minutes). For our orchid root study, we used 15 minute time intervals for cycles, but this can be customized as well as the speed of the camera movement.

To initiate imaging, we used hardware triggering onboard the camera. The FLIR Flea3 USB3 camera (model FL3-U3-120S3C-C) has a hardware trigger capability through an attached GPIO cord. When the motor stops moving the belt at a given point in the array, a signal is sent from the Arduino through the GPIO cord to trigger the camera to take an image ensuring that the camera is only capturing images when it is not moving and to maintain the same position in each imaging cycle. As images are collected, they are saved to the computer connected to the system and a script runs automatically to sort each container's images into its own folder for subsequent analysis. It is possible to use a different camera to better image certain plant roots in different environments–if this is desirable, the mount will need to be modified and the camera trigger code may need to be adapted, but the rest of the system will work as expected.

Other systems designed to systematically image plants are limited by the type of container specified as well as expensive equipment and parts [13–17]. GROOT can be customized based on experimental needs (see Fig 2) and plant types, including the ability to place obstacles such as lattices and angled plates, changing lighting settings and other modifications to container size and materials [23]. The following parameters are all customizable without requiring extra materials: the height of the rows the containers are fastened to (which allows for different container heights), the position at which the camera stops (allowing for different container widths), the length of time between each imaging cycle, the light cycle timing used to illuminate the containers, and the angle and distance of the camera with respect to the containers.

In addition to physical construction of the robot, there are a variety of customizable software options for preprocessing and analysis of the resulting imaging data. We have included code and documentation for an image preprocessing and time-lapse movie creation system we have developed. This system uses printed QR code labels to identify the individual growth vessels from the raw images (S3 File: QR Code Generation, code available at https://github.com/orgs/the-rhizodynamics-robot/repositories), which are then subsequently sorted into individual subdirectories, after which (optionally) stabilized time-lapse videos are created using the open source video editing software FFMPEG [24] [S4 File: Image Sorting, code available at https://github.com/the-rhizodynamics-robot]. This system has been tested on Ubuntu 20.04 but should be portable to other operating systems.

## Results and discussion

An initial GROOT prototype was built and used to collect images of rice root growth with a focus on the dynamic tip movement called circumnutation [23]. These datasets showed plants

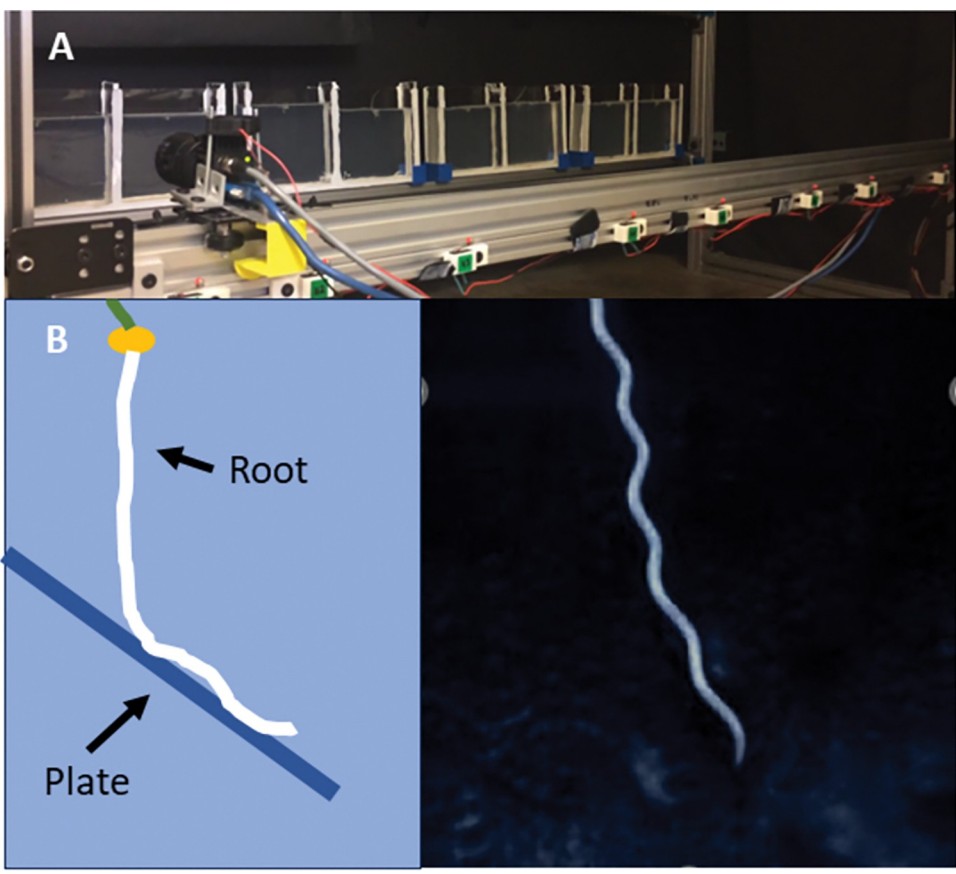

**Fig 2. Examples of customization possibilities for GROOT.** A) Quasi 2D container used for root imaging as a complement to the 3D containers used in the studies above B) Images demonstrating environmental heterogeneities that can be added to the root imaging container growth media, such as angled plates and the corresponding growth patterns that formed when growing *Oryza sativa* rice roots towards them.

growing for seven days with image acquisition every 15 minutes. After the successful deployment of the prototype, several additional systems were built at different locations. We built an identical system to image rice roots as well as four smaller versions that fit inside growth chambers, including a Shel Lab SRI20p refrigerated incubator and a Percival AR-36L2, to control temperature and humidity of the growth environment. The throughput of these smaller versions were limited by the size of the growth chambers.

To demonstrate the long-time capabilities of GROOT, an additional system was constructed at the Atlanta Botanical Gardens (ABG) (Atlanta, GA, USA) with the capability of imaging 90 experimental containers to capture time lapse imaging of orchid root growth in a climate controlled room. Orchid roots grow slowly and it can take months to see any visible growth. Thus the ability to take images of multiple orchid plants over long time periods is essential to gain insights into their root dynamics.

We were motivated scientifically to understand the differing root growth behaviors seen in terrestrial and epiphytic orchid roots. That is, orchids can grow in a wide variety of environments and climates. Some orchid roots grow in soil on the ground (terrestrial), others can grow attached to trees or rocks, with roots in the air (epiphytic) while some exhibit both types of behaviors [25]. There is limited understanding of how these roots grow in such a wide range of environments. Orchid roots also establish complex mycorrhizae relationships [26].

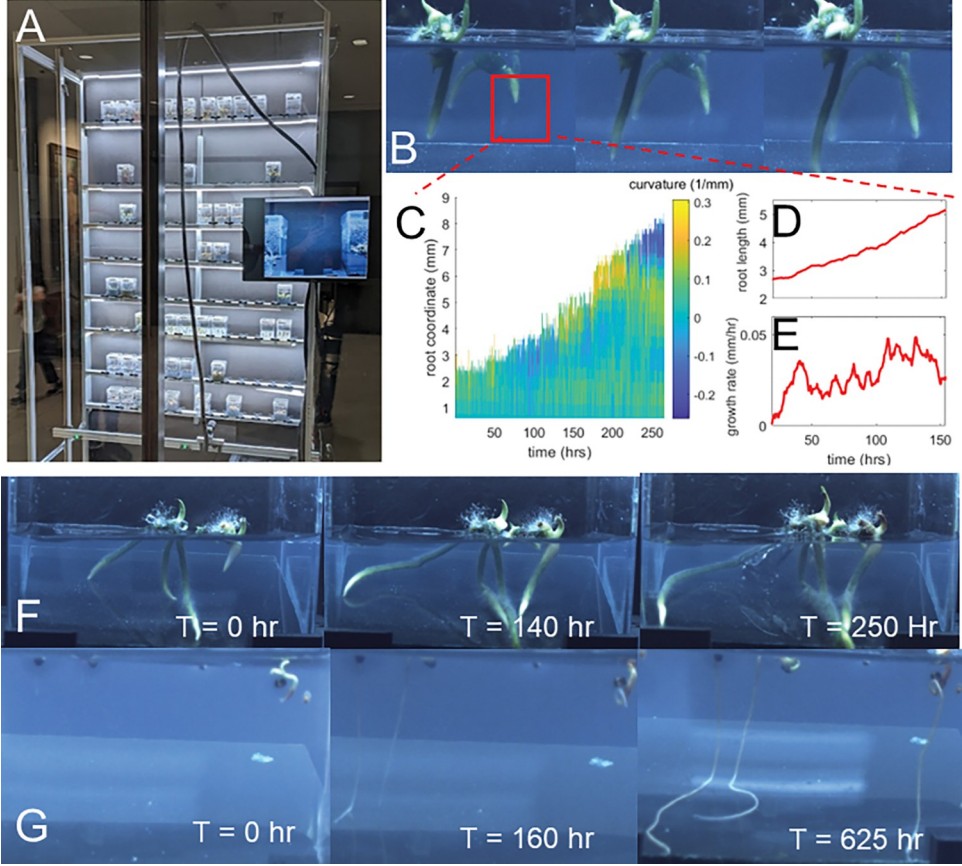

**Fig 3. Monitoring slowly growing roots using GROOT.** A) Atlanta Botanical Garden (ABG) GROOT System. B) an image series of orchid roots growing over ~100 hours, red box highlights region of interest selected for tracking. C) a heatmap of orchid root radius of curvature as a function of root length over time. Root length D) and growth rate E) over time reveal relatively constant growth over the tracked time period (~26 days). Images taken over longer time frames of F) *Platanthera integrilabia* and G) *Hudsonia montana* growth.

Understanding the dynamic growth movements of their root tips can provide insight into those relationships and help to develop novel propagation techniques for conservation purposes [27]. This system has now been used to collect image data continuously over one year, demonstrating the durability and robustness of the system.

To our knowledge, the GROOT-acquired images are the first time-lapse sequences of orchid roots growing in a gel medium. The results of the GROOT ABG experiments are shown in Fig 3. Post-capture orchid root tracking was conducted with a custom MATLAB Script. Each collected image was registered to the initial image to compensate for remaining image jitter not eliminated by the hardware sensors. Images were then cropped to a region of interest containing solely the roots of interest. The images were then binarized, creating a mask of the root profile, which was then skeletonized using morphological operations to identify the centerline along the root. This centerline was then smoothed with a spline function, allowing the calculation of quantities of interest such as root curvature (Fig 3C), root length and growth rate. Root length measurements were smoothed with a running average to reduce noise in the calculation of growth rates. Images taken over a period of ~100 hours revealed a relatively steady growth rate over time (Fig 3D and 3E). Remaining noise can be attributed to segmentation errors due to inconsistent lighting–these can be resolved with more consistent

lighting conditions and with more nuanced image binarization techniques such as subpixel localization. This is the first data collected that can establish the growth rate of different orchid species and can in turn be used to assist in development of future root growth experiments as well as for conservation purposes. For example, understanding whether orchid roots grow slower or faster under different growth conditions can be used when planning experiments to try to reestablish orchid populations.

The test of GROOT on orchids over long times allowed us to troubleshoot problems as they arose and implement changes to prevent them from recurring. These enhancements included installing a backup battery to power the system during power outages and a weekly maintenance schedule and checklist to help ensure sensors and moving parts are working correctly.

In summary we posit that the GROOT system can fundamentally change the cost, timeframe, and efficiency of studies not only of plant roots, but also across multiple disciplines in plant biology and beyond. Our proof-of-concept system has been used for three years and its design was refined to build multiple versions at different locations, which generated reproducible results. By using low cost and open-source DIY equipment, our system is designed for scientists from many different backgrounds, enabling them to construct and customize for their needs, making this a useful tool to generate high throughput image data for a wide variety of applications.

## Supporting information

**S1 File. Guide for robot assembly.**
(DOCX)

**S2 File. Guide for robot control.**
(DOCX)

**S3 File. Guide for QR code generation.**
(DOCX)

**S4 File. Guide for image sorting.**
(DOCX)

**S1 Video. Timelapse of GROOT system at the Atlanta botanical gardens.**
(MP4)

**S2 Video. Movement of imaging camera across one horizontal row and vertical movement transition to next imaging row.**
(MP4)

**S3 Video. Horizontal movement of camera and image acquisition across row on GROOT system at Georgia Tech.**
(MP4)

**S4 Video. Example of customizability of GROOT system–Non-standard imaging container and limit switch sensors are used to control LED lights for image acquisition purposes.**
(MP4)

**S5 Video. Timelapse videos of root growth created from images acquired from GROOT systems.**
(MP4)

## Author Contributions

**Conceptualization:** Isaiah W. Taylor, Erin McCaskey, Philip N. Benfey, Daniel I. Goldman.

**Data curation:** Aradhya Rajanala, Isaiah W. Taylor, Erin McCaskey, Christopher Pierce, Jason Ligon, Anupam Mijar, Milan Shah.

**Investigation:** Isaiah W. Taylor, Erin McCaskey, Christopher Pierce, Jason Ligon, Enes Aydin, Amanda Carmichael, Lauren Eserman, Emily E. D. Coffee, Philip N. Benfey, Daniel I. Goldman.

**Methodology:** Isaiah W. Taylor, Erin McCaskey, Christopher Pierce, Carrie Hunner, Amanda Carmichael, Lauren Eserman, Emily E. D. Coffee, Philip N. Benfey, Daniel I. Goldman.

**Project administration:** Daniel I. Goldman.

**Software:** Isaiah W. Taylor, Christopher Pierce, Anupam Mijar, Milan Shah.

**Supervision:** Daniel I. Goldman.

**Writing – original draft:** Aradhya Rajanala, Isaiah W. Taylor, Christopher Pierce, Philip N. Benfey, Daniel I. Goldman.

**Writing – review & editing:** Aradhya Rajanala, Isaiah W. Taylor, Christopher Pierce, Jason Ligon, Enes Aydin, Carrie Hunner, Amanda Carmichael, Lauren Eserman, Emily E. D. Coffee, Philip N. Benfey, Daniel I. Goldman.

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
