## [Decision Letter · Decision Letter 0]

14 Sep 2023

PONE-D-23-17715The rhizodynamics robot: Automated imaging system for studying long-term dynamic root growthPLOS ONE

Dear Dr. Rajanala,

Thank you for submitting your manuscript to PLOS ONE. After careful consideration, we feel that it has merit but does not fully meet PLOS ONE’s publication criteria as it currently stands. Therefore, we invite you to submit a revised version of the manuscript that addresses the points raised during the review process.

We look forward to receiving your revised manuscript.

Kind regards,

Edoardo Sinibaldi

Academic Editor

PLOS ONE

Journal Requirements:

Additional Editor Comments (if provided):

The scientific contribution is of interest, also considering the growing interest in plant-related/inspired technologies. Being aligned with the Reviewer comments, I suggest to address them in full. (In this regard and considering their relevance for the tackled subject, higher-quality images would strengthen the proposed contribution.)

Reviewers' comments:

Reviewer's Responses to Questions

**Comments to the Author**

1. Is the manuscript technically sound, and do the data support the conclusions?

Reviewer #1: Yes

2. Has the statistical analysis been performed appropriately and rigorously? 

Reviewer #1: N/A

3. Have the authors made all data underlying the findings in their manuscript fully available?

Reviewer #1: Yes

4. Is the manuscript presented in an intelligible fashion and written in standard English?

Reviewer #1: Yes

5. Review Comments to the Author

Reviewer #1: In the paper” The rhizodynamics robot: Automated imaging system for studying long-term root growth” the authors present a low-cost automatic time-lapse imaging system they have created which can be used to study different plant root systems and experimental conditions and allows the acquisition over a long time compared to the systems previously created.

I have read the paper with interest, and I believe that the system is a valuable tool for automated time-lapse acquisitions for prolonged periods of plants of different species, given the flexibility of the system. However, in my opinion, it poses some limitations in root imaging acquisition for several reasons: the image resolution seems poor; therefore, plant species with thin primary roots and lateral roots may be challenging to visualize, and the parameters that can be analyzed may be limited to thick primary roots; temperature control may also be a limitation for certain plant species given the long period of acquisition, especially for growth rate analysis. I believe it is a valuable tool, but to be used for certain types of settings and conditions, which maybe should be pointed out more.

Below I noted a few points I suggest for clarification or further improvement of the paper:

1. Please correct in the abstract the misspelling of the name of Oryza sativa and remove the capital letter in the name of the species.

2. Line 191: The authors state that they could analyze the root curvature, root length, and growth rate of the orchid roots. However, the root growth rate is affected by parameters like temperature. From what I understand, the temperature is not controlled, and it may be a system limitation for a prolonged acquisition period for certain plant species. Is there a way in which the system could be improved? You mentioned it briefly in the case of rice acquisition, but I would mention it as one of the improvements or additions that could be made to the system. Would that limit the number of samples that could be analyzed?

3. Figure 2G; in the Hudsonia montana root growth images, the roots aren’t visible. Could the images be improved or changed? in the caption, remove the capital “M” to the name Hudsonia.

4. Can the authors improve the quality of the figures presented in the article? The resolution could be better, especially Figure 3.

5. Is it possible to improve the resolution of the images? or is that in plan for a future work?

6. PLOS authors have the option to publish the peer review history of their article (what does this mean?). If published, this will include your full peer review and any attached files.

Reviewer #1: **Yes: **Marilena Ronzan

---

## [Author Response · Author response to Decision Letter 0]

30 Oct 2023

We have included a response to reviewers as a separate file during submission. We thank the editor and reviewer for their time and attention.

---

## [Decision Letter · Decision Letter 1]

30 Nov 2023

The rhizodynamics robot: Automated imaging system for studying long-term dynamic root growth

PONE-D-23-17715R1

Dear Dr. Rajanala,

We’re pleased to inform you that your manuscript has been judged scientifically suitable for publication and will be formally accepted for publication once it meets all outstanding technical requirements.

Kind regards,

Edoardo Sinibaldi

Academic Editor

PLOS ONE

Additional Editor Comments (optional):

Reviewers' comments:

Reviewer's Responses to Questions

**Comments to the Author**

1. If the authors have adequately addressed your comments raised in a previous round of review and you feel that this manuscript is now acceptable for publication, you may indicate that here to bypass the “Comments to the Author” section, enter your conflict of interest statement in the “Confidential to Editor” section, and submit your "Accept" recommendation.

Reviewer #1: All comments have been addressed

2. Is the manuscript technically sound, and do the data support the conclusions?

Reviewer #1: Yes

3. Has the statistical analysis been performed appropriately and rigorously? 

Reviewer #1: N/A

4. Have the authors made all data underlying the findings in their manuscript fully available?

Reviewer #1: Yes

5. Is the manuscript presented in an intelligible fashion and written in standard English?

Reviewer #1: Yes

6. Review Comments to the Author

Reviewer #1: Dear Authors,

All my comments have been addressed, therefore I will suggest your paper accept for publication.

7. PLOS authors have the option to publish the peer review history of their article (what does this mean?). If published, this will include your full peer review and any attached files.

Reviewer #1: **Yes: **Marilena Ronzan

---

## [Editor Report · Acceptance letter]

12 Dec 2023

PONE-D-23-17715R1 

The rhizodynamics robot: Automated imaging system for studying long-term dynamic root growth 

Dear Dr. Rajanala:

I'm pleased to inform you that your manuscript has been deemed suitable for publication in PLOS ONE. Congratulations! Your manuscript is now with our production department. 

Kind regards, 

on behalf of

Dr. Edoardo Sinibaldi 

Academic Editor

PLOS ONE